# Iodinated Contrast Medium Affects Urine Cytology Assessment: A Prospective, Single-Blind Study and Its Impact on Urological Practice

**DOI:** 10.3390/diagnostics12102483

**Published:** 2022-10-13

**Authors:** Milan Kral, Pavel Zemla, David Hradil, Hynek Skotak, Igor Hartmann, Katerina Langova, Jan Bouchal, Daniela Kurfurstova

**Affiliations:** 1Department of Urology, University Hospital Olomouc, 77900 Olomouc, Czech Republic; 2Department of Medical Biophysics, Medical Faculty, Palacký University, 77900 Olomouc, Czech Republic; 3Department of Clinical and Molecular Pathology, University Hospital Olomouc, 77900 Olomouc, Czech Republic; 4Department of Clinical and Molecular Pathology, Institute of Molecular and Translational Medicine, Medical Faculty, Palacký University, 77900 Olomouc, Czech Republic

**Keywords:** urothelial cancer, upper urinary tract, cytology, The Paris System, iodinated contrast medium

## Abstract

During endoscopic procedures for suspected urothelial tumors of the upper urinary tract, radiographic imaging using an iodinated contrast medium is often required. However, following ureteropyelography, we detected changes in cytology characteristics not correlating with real cytology findings in naive urine. The aim of our study was to assess cytology changes between naive and postcontrast urine according to The Paris System of cytology classification. **Methods:** We prospectively assessed urine samples from 89 patients (23 patients with histologically proven urothelial cancer and 66 healthy volunteers). The absence of malignancy was demonstrated by CT urography and/or ureteroscopy. The study was single blind (expert cytopathologist) and naïve Paris system for urine cytology assessment was used. Furthermore, additional cytological parameters were analyzed (e.g., specimen cellularity, degree of cytolysis, cytoplasm and nucleus color, chromatin and nucleo-cytoplasmic ratio). **Results:** Our study showed statistically significant differences when comparing naïve and postcontrast urine in healthy volunteers (only 51 % concordance, *p* = 0.001) versus malignant urine specimens (82 % concordance). The most important differences were in the shift from The Paris System category 2 (negative) to 1 (non-diagnostic) and from category 2 (negative) to 3 (atypia). Other significant changes were found in the assessment of specimen cellularity (*p* = 0.0003), degree of cytolysis (*p* = 0.001), cytoplasm color (*p* = 0.003), hyperchromasia (*p* = 0.001), course chromatin (*p* = 0.002), nucleo-cytoplasmatic ratio (*p* = 0.001) and nuclear borders’ irregularity (*p* = 0.01). **Conclusion:** Our unique study found crucial changes in the cytological assessment of naive and postcontrast urine and we confirm that postcontrast urine is more often assessed as abnormal, suspect or non-diagnostic. Therefore, before urine collection for cytology, the clinician should avoid administration of iodinated contrast into the urinary tract.

## 1. Introduction

Urothelial carcinomas are the sixth most common of all malignancies worldwide and include upper-urinary-tract tumors (calyx system and ureters) and lower-urinary-tract tumors (predominantly bladder tumors). While bladder tumors account for 90–95% of all urothelial tumors, the upper urinary tract accounts for 5–10% [1]. The basic diagnostic methods include radiological methods (ultrasonography, CT/MR urography), endoscopic examinations (ureterorenoscopy, cystoscopy) and urine cytology. It is the cytology of urine that plays an irreplaceable role in the diagnosis of subtle tumors of the urinary tract, such as carcinoma in situ. This is because urine cytology is characterized by high sensitivity and specificity in the diagnosis of high-grade urothelial carcinoma. Conversely, in the case of low-grade tumors, the specificity is still high, but the sensitivity decreases to only 25–40%. The disadvantage of the cytological evaluation of urine is that it exhibits significant interindividual and intraindividual variability. To unify the method of evaluation as well as the variability of urine cytology assessment, The Paris System for Reporting Urinary Cytology (TPS) was introduced into practice in 2016 [2] and updated in 2022 [3]. Another option is to use auxiliary techniques such as image analysis technology for cytopathological specimens [4].

Of crucial importance is not only the role of the pathologist in terms of the result, but also the method of urine collection, storage and processing. Factors that further affect the outcome of cytology include the presence of a urinary tract infection, a foreign body in the urinary tract (including urolithiasis), or previous intravesical immunotherapy for bladder cancer [5]. However, there is a lack of papers in the literature that address the consequences of the administration of an iodinated contrast agent during endoscopic examinations on cytology result according to TPS. The role of the contrast agent gains importance in examining the upper urinary tract for suspected urothelial tumors. Because retrograde ureteropyelography is often required during ureterorenoscopy, an assessment of the effect of a contrast agent on the outcome of urine cytology has shown to be important in routine clinical practice. We have repeatedly encountered a situation where patients without urothelial carcinoma of the upper urinary tract exhibited suspicious or malignant cytology in urine samples taken during ureteroscopy after previous ureteropyelography. The crutial purpose of the study was to assess the differences in the categorization and interpretation of cytology results in naive and postcontrast urine and their clinical consequences for daily practice.

## 2. Materials and Methods

### 2.1. Patient Selection and Study Blinding

Between November 2017 and May 2021, we performed a prospective analysis of urine samples from 89 randomly arriving patients. Of these, 23 patients had urothelial carcinoma in the lower urinary tract at the time of urine collection. Negative controls included 66 patients, partly healthy volunteers (21 cases), and partly patients whose urinary tract was examined for other reasons (e.g., urolithiasis, 45 cases). The absence of urothelial malignancy of the upper urinary tract was assessed by CT urography and/or ureteroscopy. In 12 cases of young health-care workers/medical students, ultrasonography of the urinary tract was performed only due to ethical reasons and due to radiation hygiene; these subjects had no history of gross haematuria and had a negative urinary sediment during the study. In all healthy control patients, a follow-up urine check-up was performed during the following 1–3 months after the study to exclude haematuria or leukocyturia. There were no relationships between researchers and students (family members or, e.g., students preparing for urology/pathology exam), nor were there relationships between students and the included patients. The study was conducted at the Department of Urology and the Department of Clinical and Molecular Pathology of the Faculty of Medicine and Dentistry, Palacky University Olomouc and the University Hospital Olomouc. The protocol and design of the study were approved by the Ethics Committee of the University Hospital Olomouc (ref. no. 62/18, signed on 11 June 2018) and each of the study participants gave written informed consent. To reduce evaluation bias, all urine samples were evaluated by a member of the team of pathologists from the Department of Clinical and Molecular Pathology. Finally, the urine specimens were reevaluated by a single board qualified cytopathologist (D.K.) to prove the final specimen description. The study was double-blinded to the pathologist: 1) the pathologist was unaware whether the urine sample was from a healthy volunteer/control or from a patient with urinary malignancy; 2) the pathologist was not informed which of the two samples from each patient contained a contrast agent and which was a lavage sample without contrast agent (or naive urine).

### 2.2. Urine Collection

At baseline, urine was collected from each patient for sediment analysis (to assess erythrocyturia and leukocyturia). The iodine contrast medium (Ultravist^®^ 370) was diluted 1:1 with saline. During the procedure/examination itself, two urine samples (approximately 20–30 mL each) were obtained before and after administration of 10 mL of contrast medium. Urine collection technique varied according to the type of procedure. In patients indicated for ureterorenoscopy, lavage cytology was carried out as part of this procedure (before and after ureteropyelography). In patients with bladder tumors and healthy controls, a contrast agent was added to the urine sample (voided spontaneously or taken perioperatively).

### 2.3. Urine Processing Procedure

Urine samples were collected in a sterile transport tube and the sample was transported to the Department of Pathology. There, the urine sample was centrifuged in an MPW centrifuge at 2500 rotations per minute (rpm) for 2 min. Using a pipette, the clear fluid was removed from the top of the tube. Five drops of saccomanno fluid were added to the concentrated contents at the bottom of the tube and mixed. The material thus prepared was processed in cytosine and centrifuged for 6 min at 1000 rpm. The prepared slides were stained using the May–Grunwald–Giemsa Romanowski method.

### 2.4. Urine Evaluation

Urine was evaluated according to The Paris System (1–non-diagnostic, 2–negative for high-grade urothelial cancer, 3–atypical cells, 4–suspected high-grade urothelial cancer, 5–high-grade urothelial cancer, 6–low-grade urothelial cancer, 7–others: primary and secondary tumors). The pathologist assessed to what extent naive and postcontrast urine samples correlated with the The Paris System. Criteria for representative material according to The Paris System classification were at least 2 urothelial cells/HPF in 10 HPF at a mangification of 400. Simultaneously, other cytological features (specimen cellularity, cell evaluability, degree of cytolysis, cytoplasm clarity, hyperchromasia, coarse chromatin, nucleo-cystoplasmic ratio and regularity of nuclear borders) were assessed. Specimen cellularity was evaluated using a three-point scale. Degree of cytolysis was evaluated using a four-point scale, and four parameters (hyperchromasia, coarse chromatin, nucleo-cytoplasmic ratio, and irregular nuclear borders) were evaluated using a two-point scale (see Table 1). In addition to these cytological parameters, we also focused on whether leukocyturia was present and whether its extent could affect the cytological outcome and final cytological categorization.

### 2.5. Statistical Analysis 

The McNemar–Bowker test was used for statistical comparison of The Paris System while other cytological parameters were verified by the Wilcoxon test. Leukocyturia was evaluated by the Kruskal–Wallis test. All tests were performed at the level of statistical significance of 0.05. IBM SPSS Statistics for Windows, Version 23.0 statistical software was used for statistical processing. Armonk, NY: IBM Corp. We could not estimate the optimal sample size before starting the study due to the nature of the study, as we did not find a pre-existing study on this topic with the same study design. For this reason, it was not possible to determine preliminary results and perform a power calculation.

## 3. Results

Demographic data of the cohort are shown in Table 2. Although cases with low-grade urothelial carcinoma (diagnosed by transurethral resection) were present in the group of patients, cytologically, TPS category 6 samples, i.e., low-grade urothelial cancer changes, were not identified. Similarly, there were no cases of TPS category 7 (other tumors). For this reason, samples of categories 4–7 were included in a common group. There was no TPS category 2 patient (negative for high-grade urothelial cancer) in the group of patients with proven urothelial carcinomas. The division of naive urine into categories according to The Paris System is shown in Table 3. A comparison of urinary findings from the upper urinary tract and the bladder shows that atypical findings, i.e., category 3, occur significantly more in the upper urinary tract (*p* = 0.002, bold print in Table 3).

We evaluated the cohort as a whole (i.e., samples from patients with proven urinary tract malignancies + samples from healthy volunteers) and separately from patients with urothelial tumors and healthy volunteers. When analyzing the whole cohort with the McNemar–Bowker test, categorization according to The Paris System was shown to be different in naive and in postcontrast urine samples. We found this difference to be statistically significant (*p* = 0.0002). In 52 samples (58 %, bold print in the table), the categorization was identical, but in 37 samples (42%), there were differences in the categorization. The most common differences were as follows: 10 (11%) samples were evaluated as 2 and 1 in naive and postcontrast urine, respectively; 10 (11%) samples were evaluated as 2 and 3 in naive and postcontrast urine, respectively (see Table 4).

A subanalysis of the cohort failed to show a statistically significant difference when comparing naive and postcontrast urine in patients with urothelial carcinomas, with the obtained urine values agreeing in 19 out of 23 patients (i.e., 83%, *p* = 0.135). This means that the addition of a contrast agent to the cancer urine sample did not lead to significant changes in cytology categorization. However, fundamental differences were evident in the samples of healthy controls. 

The McNemar–Bowker test showed that the evaluation of control group samples based on the The Paris System was different for naive and postcontrast urine. Compliance was achieved in only 33 out of 66 urine samples (i.e., in 50%). The most significant changes were shifts across categories: from category 2 (negative for high-grade urothelial cancer) to category 1 (non-diagnostic) in 11% of cases and from category 2 (negative for high-grade urothelial cancer) to 3 (atypia) in 11% of cases. See Figure 1 and Figure 2 for changes in the cytological smears of non-cancer patients.

In addition to changes in the categorization according to The Paris System, other significant changes in cytological subparameters were demonstrated by the Wilcoxon test (note: samples from all patients were not evaluable in all categories). We showed a statistically significant difference between naive urine and postcontrast urine in the specimen cellularity (*p* = 0.0003). Cellularity was lower in specimens with a contrast agent (38% of the specimens). Half of the specimens had the same cellularity, and in only 13% it was higher in postcontrast urine. We believe that this could be explained by the dilution of the urine sample with a contrast agent. The Wilcoxon test also showed a statistically significant difference between naive urine and postcontrast urine in cell evaluability, with agreement in 57% of cases, but, surprisingly, in as much as 30%, the clarity was lower in naive than in postcontrast urine (*p* = 0.013). In terms of cytolysis, although the agreement between samples was as much as 61%, in naive urine samples, a low degree of cytolysis was significantly more frequent than in postcontrast samples (31% vs. 8%; *p* = 0.001). Both cytoplasm and nucleus clarity showed a concordance of 66% and 63%, respectively; however, the postcontrast samples were evidently darker than the naive ones (*p* = 0.003 and 0.001, respectively)–see Table 5. The subanalysis again showed that the most significant changes occurred in the control group, which has a major clinical impact.

In the monitored cytological parameters, i.e., coarse chromatin, nucleo-cytoplasmic ratio and irregular boarders, no differences were found between naive and post-contrast material in 67%, 69% and 71%, respectively. Differences in naive samples vs. post-contrast cases were recorded in 33%, 31% and 29%, where most of these changes were due to the inconclusive material of the post-contrast specimen.

There was a secondary focus on the presence of leukocyturia and categorization according to The Paris System. It is important to mention that all patients were asymptomatic and, in case of leukocyturia, a negative urine culture test was obtainedbefore enrolling in the study (even in cases with a significant level of pyuria). Evaluation of urinary sediment samples (erythrocyturia and leukocyturia) and native cytological findings showed that samples classified as atypia had a significantly higher number of leukocytes than other samples (*p* = 0.047, Kruskal–Wallis test). In other words, with increasing leukocyturia, the accuracy of cytological evaluation decreases, and the pathologist obtains atypical findings more often (see Figure 3). The degree of erythrocyturia is not involved (Table 6).

## 4. Discussion

Urine cytology is one a crucial examination when identifying malignant diseases of the lower and upper urinary tract [1,6]. However, there are several circumstances that can lead to misinterpretation of the result of a cytological examination, affecting the clinical evaluation and following treatment of the patient. The repeated presence of abnormal cytology findings after the perioperative application of contrast material to the urinary system during ureteroscopy led us to create the design of this study. As the above results show, we confirmed the initial hypothesis that postcontrast urine showed major cytological changes, especially in healthy patients (only 51 % concordance between naïve and postcontrast urine, *p* = 0.001). Conversely, post-contrast urine cytology in patients with urothelial carcinomas did not significantly differ from naive urine cytology (82 % concordance). An increase in the N/C ratio of the present cells is one of the main diagnostic features in the sample examination algorithm. In our work, we demonstrate, using McNemar’s symmetry test, a statistically significant difference between the naïve and postcontrast N/C ratio (*p* = 0.001). No changes were found in 59 (69%) samples. In 27 (29%) samples, increased N/C shift was observed in postcontrast samples (7 samples). In other samples, it was not possible to assess these changes. Due to this fact, it is of the utmost importance to not perform urine cytology assessment after the application of a previous iodine contrast agent (e.g., retrograde pyelography).

Papanicolaou demonstrated the benefits of urine cytology as early as 1947, and since then, it has been an integral part of the diagnosiss of urothelial cancer [7,8]. Cytological samples were evaluated according to The Paris System version 1.0 (2). The second edition of this classification system [3] was introduced this year. Among others, it represents a new algorithm in the diagnosis of urinary tract tumors and focuses more on the issue of the upper urinary tract. Although The Paris System recommends Papanicolaou staining for urine specimens, our specimens were stained with Giemsa–Romanowski [9]. Giemsa–Romanowski has very good fixing properties, enables the preparation of slides from a small amount of cellular material and demonstrates an improved clarity of cells in cell clusters. However, the assessment of nuclear chromatin is worse than with Papanicolaou staining. Giemsa–Romanowski staining is one of the basic recommended stains, which we use in our routine practice, and we have many years of good experience with this.

The criterion for an evaluable cytological specimen (according to The Paris System) is the presence of at least two cells per high-power field (HPF) in 10 evaluated HPFs for enough epithelial cells to be present. However, in association with the collection, processing, and interpretation of findings, several problematic circumstances arise that reduce the yield. Although urine cytology exhibits high specificity and sensitivity in high-grade tumors, it fails to achieve similar results in low-grade (LG) tumors [7,10]. The detection and recognition of tumor cells in LG tumors is due to cell cohesiveness and the absence of nuclear atypia. Although LG tumors are more prevalent, their biological potential, i.e., the risk of local progression, is logically lower than that in high-grade tumors. Therefore, the whole concept of adjusting cytological classification is focused on the detection of high-grade tumors as well as on the standardization of findings to reduce the interindividual variability of evaluation [2,11,12]. In the case of lower-urinary-tract tumors, cystoscopy is the key diagnostic procedure, which usually detects an exophytic tumor, with cytology playing a more supportive role. However, the importance of cytology significantly increases during surveillance after the curative transurethral resection of bladder tumor, as well as after radical cystectomy and urinary tract diversion. Diagnosis of an uncertain or suspected upper-urinary-tract tumor on imaging (CT/MR urography) may be supported by evidence of high-grade cancer in cytology, especially when imaging may not detect urinary tract abnormalities in a small tumor.

A cell sample is typically obtained from a routine voided cytology specimen; at other times, instrumented urine or brush cytology are used. In a sample obtained in this way, cells from the deeper layers of the epithelium (intermediate or basal cells) are typically present [13]. For a sample to be valid in terms of epithelial cellularity, a urine volume above 30 mL is usually sufficient [14]. In clinical practice, however, we may encounter a poor interpretation of cytology findings, e.g., in massive pyuria (a competing urinary tract infection, a foreign body such as a stent, urinary catheter, or urolithiasis). This is in accordance with our conclusions, when especially severe pyuria and erythrocyturia reduced the yield and, specifically, leukocyturia led to a higher rate of categorization 3 (atypia) (see Figure 3). When epithelial cells overlap with leukocytes, they may be difficult to identify. It such a case, it is recommended to treat the urinary tract infection first and repeat the urine sampling after that. In this respect, our results differ from those of Wojcik, which showed no effect of urinary tract infection or urolithiasis on the cytological character of the finding (patients were classified as 2 in the Wojcik’s study, i.e., negative for high-grade urothelial carcinoma) [15]. Therefore, we examined urinary sediment and cytology at the same time, and in the presence of leukocyturia, we searched for urinary tract infection, provided targeted antibiotic therapy when bacteriuria was detected, and only subsequently examined urine cytology.

Our second recommendation is to reduce single observer bias by having at least two or more observers review abnormal/inconclusive cases. It is quite logical that the role of pathologists in the processing and analysis of cytological samples is crucial. It must be considered that the result of cytology is significantly affected by interindividual variability, as has been repeatedly demonstrated [2,11,16,17]. Accordingly, a significant strength of our study is that the whole cohort was analyzed and re-evaluated by a single cytopathologist. Another strength of our study is that the cytologist was double-blinded to minimize the effect of previously known patient´s history or histology on the definitive evaluation of cytology. As with prostate cancer, an indicator of the pathologist’s expertise in the cytological evaluation is the frequency of atypical findings in histological/cytological evaluation. In agreement with other works, in our group of naive urine specimens, atypia was found in samples from the upper urinary tract to a greater extent than in samples from the bladder (19.6 vs. 7 %, *p* = 0.002). The rate of atypia in naive urine in our whole cohort was 13.5% (12 of 89 patients), which is less in comparison to other studies. Long, for instance, reported a frequency of cytological atypia in 22% [11] and Rosenthal reported 26% [18,19], while we are close to the results of Muus Ubago, who showed cytological atypia in 8% [10]. Importantly, the incidence of atypical cell counts in our whole postcontrast urine cohort increased to 19% (17 of 89 patients). A more detailed analysis showed that 58% changed from the category negative for high-grade cancer to atypia, and in 17%, there was a shift from an atypical finding to suspected high-grade cancer, which is of major clinical impact. Indeed, the cytological determination of atypia in the urine of a particular patient has been shown to carry an 8.3–27.5% risk of detecting high-grade urothelial cancer in the subsequent follow-up period [20]. This category of patients may benefit the most from very close follow-up, as upper-urinary-tract tumors may most threaten patients´ lives. Furthermore, atypical findings or suspicion of high-grade cancer (after iodine contrast medium application) could have a negative impact on the patient’s mental state and confidence in the healthcare professional.

It is necessary to mention certain limitations of our study. One is the fact that hospitals and laboratories usually use Papanicolaou staining for cytological specimens, while we used Giemsa–Romanowski staining during the study. This is due to the long-term tradition, the financial issues of using Papanicolaou staining and the fact that many laboratories still use Giemsa–Romanowski staining for various reasons. Another limitation of our study includes the fact that the cohort consisted of 89 patients. Not all participants were consecutive patients during that period. Although this was not an intentional selection bias, only a few endoscopic surgeons participated in the study, and they included every potentially suitable patient. Patients who disagreed to the study or patients from whom it was not possible to collect urine for cytology and send for processing in time due to operational reasons were excluded from the study. Although our results were statistically significant and both groups were represented by numerous patients, it would be advisable to repeat the study on an even larger group of participants. A limitation of the sample evaluation may be the inflammatory environment in the urine (leukocyturia) and erythrocyturia. Such a patient needs a complex urinary tract assessment. However, atypical or suspected findings do not only play an important role in urine cytology. The problem of inconclusive or non-diagnostic category is that it does not lead us closer to a precise diagnosis. Despite this, we have no clear explanation for the fact that 11% of patients from our group had a shift to the non-diagnostic category in postcontrast cytology. We suppose that the specimens were affected by cytolysis, or their cellularity did not fulfil the criteria for representative material according to The Paris System classification (as mentioned above). What was surprising in our cohort was the effect of added iodinated contrast agent on cell clarity, which was significantly superior in postcontrast urine samples. It is possible that a lower specimen cellularity may contribute to this phenomenon, as it was also evident in postcontrast samples (see Table 5).

During our study, we focused on the effect of an iodine contrast agent administered perioperatively into the urinary system or added artificially to the urine. The effect of contrast medium applicated intravenously during CT urography on urine cytology was not assessed because such the consequences are far from common practice. The time of taking cytology usually differs from the time of CT urography. We tried to make the design of the study as close as possible to the real situation, when it is often necessary to perform retrograde ureteropyelography during ureteroscopy to clarify the course of the ureter or rule out its injury. With the help of our results, we want to highlight that urine cytology sampling should never be performed after retrograde pyelography.

## 5. Conclusions

Based on our work, a hypothesis of a possible influence of iodine contrast medium on urine epithelial cells is formed. Our study showed fundamental changes in the cytological evaluation of naive urine and urine after the administration of a contrast agent into the urinary tract. Because of its design (prospective, blinded to the pathologist), this study is unique. The results confirm the assumption that postcontrast urine is more often evaluated as being cytologically abnormal, suspected of urothelial carcinoma or, conversely, non-diagnostic. Therefore, the administration of a contrast agent into the urinary tract should be avoided before collecting urine.

## Figures and Tables

**Figure 1 diagnostics-12-02483-f001:**
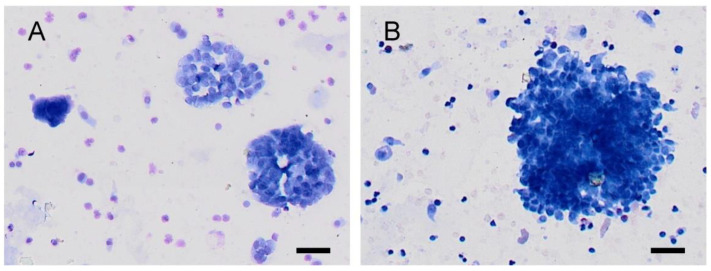
Urine cytology from a healthy volunteer (category shift from TPS2 to TPS3). (**A**): In naive urine (TPS2-negative for high-grade urothelial cancer), the cells are better seen, with uniform nuclei and fine chromatin. (**B**): In the postcontrast urine (TPS3–atypia), the visibility of cell clusters is clearly worse. Nuclei appear hyperchromic, and in some cells, notches in the karyomembrane are also visible. A scale bar represents 50 µm.

**Figure 2 diagnostics-12-02483-f002:**
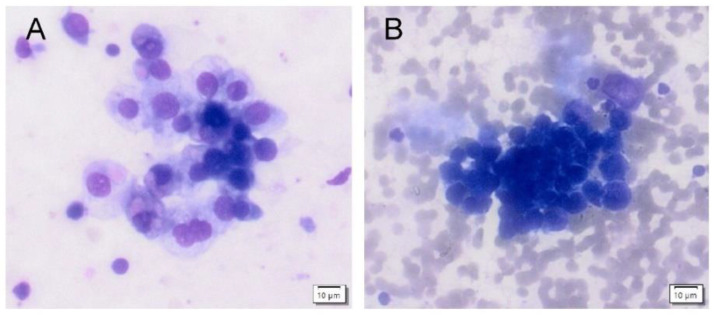
Urine cytology from a healthy volunteer (category shift from TPS3 to TPS5). (**A**): High cellularity specimen from an naive urine (TPS3-atypia) with cohesive clusters of atypic urothelial cells with slightly enlarged nuclei are present. The chromatin is fine, and the nuclear membrane is smooth. (**B**): High cellularity specimen from a postcontrast urine (TPS5–high-grade urothelial cancer) with numerous erythrocytes in the backround. Fewer distinct clusters of small cells are present with enlarged hyperchromatic nuclei with clumped chromatin and irregular nuclear borders. Nuclei are enlarged with higher nucleo-cytoplasmic ratio.

**Figure 3 diagnostics-12-02483-f003:**
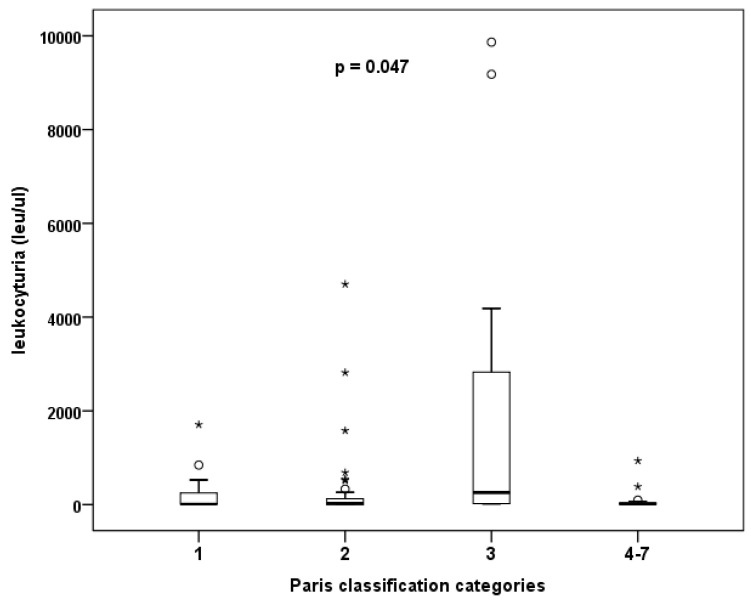
The effect of degree of leukocyturia on the frequency of atypical findings in cytology.

**Table 1 diagnostics-12-02483-t001:** Evaluation of other cytological parameters.

	1	2	3	4
specimen cellularity	low	medium	high	x
cell evaluability	good	medium	low	unevaluable
degree of cytolysis	low	medium	high	complete cytolysis
cytoplasm clarity	light	middle	dark	
hyperchromasia	not-present	present		
coarse chromatin	not-present	present		
increase nucleo-cytoplasmic ratio	not-present	present		
irregular nuclear borders	not-present	present		

**Table 2 diagnostics-12-02483-t002:** Demographic data and patient characteristics.

No. of Participants	Total	89 pts	
Sex	male	58 pts (65%)	
	female	31 pts (35%)	
Age	range	23–92 years	median 61 years
Patient category	urothelial cancer patients	23 pts (26%)	(17 high grade, 6 low grade)
	patients with no history of urothelial cancer	21 pts (24%)	healthy volunteers
		45 pts (50%)	lithiatic patients/other
Type of cytology	spontaneously voided urine	23 pts (26%)	
	washing cytology (bladder/upper tract)	66 pts (74%)	
Source of urine	upper urinary tract	46 pts (52%)	
	bladder	43 pts (48 %)	

**Table 3 diagnostics-12-02483-t003:** Distribution of naive urine samples into The Paris System categories.

Fisher’s Exact Test: *p* = 0.002			The Paris System Categories (Naive Urine)				Total
			Non-Diagn (1)	Neg * (2)	Atypia (3)	Tumors ** (4–7)	
	kidney/ureter	Count	7	24	9	6	46
		%	15.2%	52.2%	19.6%	13.0%	100.0%
	urinary bladder	Count	6	13	3	21	43
		%	14.0%	30.2%	7.0%	48.8%	100.0%
Total		Count	13	37	12	27	89
		%	14.6%	41.6%	13.5%	30.3%	100.0%

* Negative for HG cancer, ** Suspect HGUC, HGUC, LGUC and other tumours. Abbreviations: non-diagn, non-diagnostic; HG, high grade; HG UC, high-grade urothelial cancer; LG UC, low-grade urothelial cancer.

**Table 4 diagnostics-12-02483-t004:** Crosstabulation of naive and postcontrast urine specimens according to The Paris System.

McNemar-Bowker Test, *p* = 0.0002			The Paris System Categories (Postcontrast)				Total
			Non-Diagn (1)	Neg * (2)	Atypia (3)	Tumors ** (4–7)	
Paris classification (naive)	1	Count	10	1	2	0	13
		% of Total	11.2%	1.1%	2.2%	0.0%	14.6%
	2	Count	10	17	10	0	37
		% of Total	11.2%	19.1%	11.2%	0.0%	41.6%
	3	Count	5	0	4	3	12
		% of Total	5.6%	0.0%	4.5%	3.4%	13.5%
	4–7	Count	5	0	1	21	27
		% of Total	5.6%	0.0%	1.1%	23.6%	30.3%
Total		Count	30	18	17	24	89
		% of Total	33.7%	20.2%	19.1%	27.0%	100.0%

* Negative for HG cancer, ** Suspect HGUC, HGUC, LGUC and other tumours. Abbreviations: see Table 3.

**Table 5 diagnostics-12-02483-t005:** Evaluation of cytological parameters.

Parameter Evaluated	Postcontrast Urine Compared to Naive Urine	Concordance between Naive/Postcontrast Urine	Significance
specimen cellularity	lower	50%	*p* = 0.0003
cell evaluability	higher	57%	*p* = 0.013
degree of cytolysis	higher	61%	*p* = 0.001
cytoplasm clarity	lower	66%	*p* = 0.003
hyperchromasia	lower	63%	*p* = 0.001
coarse chromatin	lower	67%	*p* = 0.002
nucleo-cytoplasmic ratio	higher	69%	*p* = 0.001
irregular nuclear borders	lower	71%	*p* = 0.010

**Table 6 diagnostics-12-02483-t006:** Effect of erythrocyturia and leukocyturia on The Paris System.

	The Paris System Category				
Total Group	1 (*n* = 13)	2 (*n* = 37)	3 (*n* = 12)	4–7 (*n* = 27)	
	Median (Min/Max)	Median (Min/Max)	Median (Min/Max)	Median (Min/Max)	*p* Value
Erythrocyturia *	9 (0/9200)	134 (0/8523)	46 (3/7019)	56 (0/18,898)	0.798
Leukocyturia *	12 (0/1706)	29 (0/4703)	257 (16/9866)	18 (0/936)	0.047

* cells per µL.

## Data Availability

Not applicable.

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
