# Peer review of "Iodinated Contrast Medium Affects Urine Cytology Assessment: A Prospective, Single-Blind Study and Its Impact on Urological Practice"

_diagnostics, 2022, doi:10.3390/diagnostics12102483_

Round 1

Reviewer 1 Report

Thank you for the opportunity to review this interesting paper. In general, it seems to have a clear background with coherent progress through-out the manuscript. The English language is ok, but would definitely benefit from professional editing.

Introduction: adequate (although will be better with improved language).

Materials and Methods: It was not clear whether power calculations were performed to deduce whether this cohort was sufficient for significance in findings. Please clarify.

Results: Adequate. Line 175-176 states that the evaluation of control samples was different for naïve and post-contrast urine. How can the evaluation be different, shouldn’t they be evaluated in the same way? Or is the implication that the result was different? It becomes clearer by the figures, but the text is not transparent here. Please clarify.

Discussion: Can be improved. This is maybe silly, but it is preferable to write at least the start of the Discussion so, that it can stand independently. So do not start with referring to what has been stated in some earlier paragraph, rather lift up your result clearly. Many readers try to find the take home by reading the last paragraph of the Introduction to see what has been attempted and then the first paragraph of the Discussion to see what was achieved. Especially in this manuscript, where there is a clear result, please lift it up in the first paragraph clearly. Do not begin a paragraph with the word “But”. Please add a section with limitations to the Discussion.

The message from this paper is to take cytology during cystoscopy prior to upper tract contrast studies for evaluation of upper tract pathologies. I think this is customary to have most clinicians do, but this paper lifts up a possible confounding factor if so is not done. I wish the Discussion would lift the message with better clarity and impact.

Reviewer 2 Report

"Iodinated Contrast Medium Affects Urine Cytology Assessment: A Prospective, Single-Blind Study and Its Impact for Urological Practice" has revealed interesting results especially for urologists.

After this article, physicians showed that contrast agent should not be used when collecting urine or urological tissue samples.

Reviewer 3 Report

This study examined the correlation between contrast media and pathological diagnosis. Their hypothesis is based on their clinical experience and interesting. However, there are some concerns to address. 

1. They chose 89 randomly arriving patients. I wonder their methods have selection bias. Ideally, all patients should be included during the period. Please explain and discuss about this bias.

2.  Twelve young volunteers were included as controls. It is strange. Especially, their samples were collected by spot urine and added contrast media afterwards. Sample correction methods were different and have to be evaluated separately. Have you evaluated controls without these volunteers? I recommend evaluating in three groups. Please discuss.

3. I completely agree with their conclusions, contrast media affects pathological diagnosis underestimated by cytolysis. My question is whether intravenous contrast media (ex., enhanced CT) affects cytology, as well as through catheter?

4. Please show clinical trial number approved by Ethics Committee of their institute because their study includes normal volunteers, especially medical students. Please show there are no hierarchical relationships between them. 

Author Response

Response to Reviewers

Reviewer 3

Comments and Suggestions for Authors

This study examined the correlation between contrast media and pathological diagnosis. Their hypothesis is based on their clinical experience and interesting. However, there are some concerns to address. 

They chose 89 randomly arriving patients. I wonder their methods have selection bias. Ideally, all patients should be included during the period. Please explain and discuss about this bias.

Thank you for the positive evaluation and for pointing out this issue. We have added the following text into the novel paragraph with limitations of our study: “Another limitation of the study was that not all participants were consecutively coming patients during that period. Although this was not an intentional selection bias, the reason was that only a few endoscopic surgeons participated in the study, and they included every potentially suitable patient. Patients who disagreed to the study or patients from whom it was not possible to collect urine for cytology and send it for processing in time due to operational reasons were excluded from the study.”

  1. Twelve young volunteers were included as controls. It is strange. Especially, their samples were collected by spot urine and added contrast media afterwards. Sample correction methods were different and have to be evaluated separately. Have you evaluated controls without these volunteers? I recommend evaluating in three groups. Please discuss.

Thank you for Your concern. It is ethically inadmissible to perform a different method of collecting urine samples from a healthy volunteer than the above mentioned. However, we do not perceive a fundamental difference. The explanation is that when the contrast agent was applied during the ureteroscopy, it was administered very slowly to avoid complications (intrarenal reflux and related e.g., septic complications) and the urine with the administered contrast agent was subsequently collected. Thus, the contact time of urine/epithelia with the contrast material is practically identical in both healthy volunteers and operated patients. After consulting with the statistician, we did not create a third group.

  1. I completely agree with their conclusions, contrast media affects pathological diagnosis underestimated by cytolysis. My question is whether intravenous contrast media (ex., enhanced CT) affects cytology, as well as through catheter?

Thank you very much for Your comment. The following text has been added at the end of Discussion: „During our study, we focused on the effect of a iodine contrast agent administered perioperatively into the urinary system or added arteficially to the urine. The effect of contrast medium applicated intravenously during CT urography on urine cytology was not assessed because such consequence is far from common practice. The time of taking cytology usually differs from the time of CT urography. Therefore, the time of taking urine sample is not connected to the time of CT examination. We tried to make the design of the study as close as possible to the real situation, when it is often necessary to perform retrograde ureteropyelography during ureteroscopy to clarify the course of the ureter or rule out its injury. With the help of our results, we want to highlight that urine cytology sampling after retrograde pyelography should never be performed.”

  1. Please show clinical trial number approved by Ethics Committee of their institute because their study includes normal volunteers, especially medical students. Please show there are no hierarchical relationships between them. 

The reference number of the Ethics Committee approval has been added to the Methods - ref. no. 62/18, signed on 11th June 2018. The original document has previously been provided during the manuscript submission.

The following text has also been added to the Methods: „There were no relationships neither between researchers and students (family members or e.g. students preparing for urology/pathology exam) nor relationships between students and included patients.“

Round 2

Reviewer 3 Report

The authors have revised accordingly.